# A Computationally Efficient and Virtualization-Free Two-Dimensional DOA Estimation Method for Nested Planar Array: RD-Root-MUSIC Algorithm

**DOI:** 10.3390/s22145220

**Published:** 2022-07-13

**Authors:** Shengxinlai Han, Xin Lai, Yu Zhang, Xiaofei Zhang

**Affiliations:** College of Electronic Information Engineering, Nanjing University of Aeronautics and Astronautics, Nanjing 211106, China; laixin@nuaa.edu.cn (X.L.); yyzz@nuaa.edu.cn (Y.Z.); zhangxiaofei@nuaa.edu.cn (X.Z.)

**Keywords:** nested planar array, direction of arrival estimation, degree of freedom, reduced-dimensional, polynomial root finding

## Abstract

To address the problem of expensive computation in traditional two-dimensional (2D) direction of arrival (DOA) estimation, in this paper, we propose a 2D DOA estimation method based on a reduced dimension and root-finding MUSIC algorithm for nested planar arrays (NPAs). Specifically, the algorithm proposed in this paper transforms the problem based on 2D spectral peak search into two one-dimensional estimation problems by reducing the dimension, and then transforms the one-dimensional estimation problem into a problem of polynomial root finding. Finally the parameters are paired to realize the 2D DOA estimation. The proposed algorithm not only performs two root finding operations directly according to the 2D spectral function transformation, avoiding the performance degradation caused by intermediate operations, but can also fully exploit the enlarged array aperture offered by NPAs with reduced computational complexity and no need for virtualization. The superiorities of the proposed algorithm in terms of estimation accuracy and complexity are verified by simulations.

## 1. Introduction

Two-dimensional direction of arrival (2D-DOA) estimation has attracted extensive attention over the years and has been applied to wireless communication, sonar, radar, and many other fields [1,2,3]. There have been increasing number of algorithms proposed for two-dimensional DOA estimation such as estimation of signal parameters via rotational invariance techniques (ESPRIT) [4,5,6], multiple signal classification (MUSIC) [7], parallel factor (PARAFAC), and the propagator method (PM) [8] technique based on various array geometries, for example, L-shaped arrays [9,10], two parallel linear arrays [11], uniform planar arrays (UPA) [12,13,14], and uniform circular arrays [4]. In addition, traditional 2D-DOA methods mainly include spectral peak search [7,15,16,17,18,19] and rotation invariance algorithm [4,5,6,7,8]. Among them, although the 2D MUSIC algorithm based on spectral peak search [7] has high DOA estimation accuracy, 2D total spectrum search (TSS) has great computational cost. Rotation invariance algorithm such as 2D ESPRIT can directly calculate DOA estimation, avoiding the complex calculation of 2D spectral peak search, but cannot achieve the estimation performance of spectral peak search. Of course, there are many new DOA estimation methods based on information geometry [20,21,22], which will not be discussed in this paper. Ref. [23] proposes a MUSIC-based dimensionality reduction and root-finding algorithm for uniform planar arrays (RD-ROOT-UPA), which further reduces the amount of computation by reducing the dimension of the spectral function and then finding the root. However, in order to avoid spatial aliasing, these traditional arrays require adjacent elements to be no more than half a wavelength apart, which induces a heavy mutual coupling effect at the same time. Moreover, the limited array aperture degrades the DOA estimation performance [24,25].

To solve the above problems, various sparse arrays have been proposed, such as minimum redundant arrays [26], coprime arrays [24,27,28] and nested arrays [29]. Compared with traditional arrays, sparse arrays have the advantages of expanded array aperture, increased degrees of freedom (DOF), and lower mutual coupling [30]. In Ref. [15], the 2D-MUSIC algorithm is used to calculate the coprime planar array (CPA), and a two-dimensional partial spectrum search (2D-PSS) method is further proposed to reduce the calculation burden of 2D-TSS. Combining the reduced dimension MUSIC (RD-MUSIC) [16] algorithm with the PSS algorithm has been attempted in order to further reduce complexity by performing reduced-dimension transformation [17]. The above methods [15,17,31] belong to the decomposition algorithm, which handles the data received by each subarray separately and then puts the estimates of each subarray together to confirm the final DOA. Unfortunately, the mutual information of the two subarrays is ignored. In order to make the best of all information, Ref. [32] superimposes and jointly processes the outputs of the two subarrays to avoid ambiguity, but the two-dimensional spectrum search still brings a heavy computational burden. In order to make full use of all the information of the received signal of the coprime planar array (CPA) while reducing the computational complexity, a method similar to that in Ref. [23] was applied to the CPA [33], with the CPA regarded as an array extracted from its corresponding full array; this method processes the received signals of the two subarrays jointly to obtain better estimation results. However, the two subarrays of the CPA are arranged in the manner of a common origin, and the array aperture may be further enlarged if they are connected in series.

Inspired by the above methods, in order to further expand the array aperture, we propose a reduced dimension root finding algorithm for nested planar arrays (NPAs). Specifically, we first calculate the noise subspace according to the covariance matrix of the received signal to obtain the 2D spatial spectral function. Then, we separate two variables in the 2D spectral function to realize dimension reduction, and the roots of the polynomials obtained by dimensionality reduction are found respectively. This transforms the 2D spectral search into two 1D polynomial root finding operations, which greatly reduces the huge amount of computation brought by the spectral peak search. Finally, after two parameters are paired, the DOA estimation results can be calculated. Due to the existence of adjacent elements with half wavelength spacing in NPA, the ambiguity-free characteristic of NPAs can avoid additional disambiguation operations, and the proposed method can also refrain from virtualizing the NPA.

We summarize the major contributions of our work as follows:We use NPAs instead of UPAs and CPAs to further expand the array aperture and obtain better estimation results;We use the reduced dimension and root-finding MUSIC algorithm for 2D DOA estimation. The proposed algorithm changes the 2D spectral search into two 1D estimation problems and then transforms the 1D estimation problem into a root finding problem of a 1D polynomial, which reduces the complexity while maintaining the estimation accuracy;We regard the nested array as extracted from a uniform array with the same array aperture and use the polynomial root finding method to deal with the problem, avoiding the virtualization process of NPA.

We summarize this paper below. Section 2 recommends the data model of NPA. The proposed algorithm is expounded in Section 3 and Section 4, which analyze the complexity and DOFs of the proposed algorithm. Section 5 gives simulation results to verify the efficiency of the proposed algorithm, and Section 6 is a conclusion of the whole paper.

### Notations

Bold uppercase characters denote matrices, while the lowercase characters mean vectors. ${\left(\cdot \right)}^{T}$ and ${\left(\cdot \right)}^{H}$ represent the transpose and conjugate transpose; ${\left(\cdot \right)}^{*}$ and ${\left(\cdot \right)}^{-1}$ denote conjugate operation and inverse, respectively. ⊙ and ⊗ are the Khatri–Rao product and Kronecker product, respectively. angle(⋅) represents the phase operator. Rank(⋅) is the rank of the matrix. det(⋅)
means the determinant of the matrix.

## 2. Data Model

Consider an NPA, which is a 2D sparse planar array extended configuration of a two-level nested linear array (NLA) with $M_{1}$ and $M_{2}$. In other words, each row and column of the NPA is a two-level NLA, which is composed of two uniform linear arrays in series. The element spacing of the first level uniform linear array is $d_{1}=\lambda /2$ with $M_{1}$ elements, while that of the second level is $d_{2}=(M_{1}+1)\lambda /2$ with $M_{2}$ elements, where λ is the wavelength. Thus, the total number of array sensors is $T={\left(M_{1}+M_{2}\right)}^{2}$.

Specifically, the location set of sensors in the NPA can be represented by $\Omega_{NPA}=\left\{\Omega_{x},\Omega_{y}\right\}$, where $\Omega_{x}=m_{1}d_{1}\cup M_{1}d_{1}+m_{2}d_{2}$, $\Omega_{y}=m_{1}d_{1}\cup M_{1}d_{1}+m_{2}d_{2}$, $0\leq m_{1}\leq M_{1}-1$, $0\leq m_{2}\leq M_{2}-1$, and $m_{1},m_{2}\in Z$
Figure 1 displays the structural layout of an NPA, taking $M_{1}=2$, $M_{2}=3$, T=25 as an example.

Suppose K narrowband far-field uncorrelated signals impact the NPA. The K signals are $(\theta_{k},\phi_{k}),k=1,2,\cdots ,K$, where $\phi_{k}$ and $\theta_{k}$ are the azimuth and elevation angles of the kth signal, respectively. Define $u_{k}=\sin\theta_{k}\sin\phi_{k}$ and $v_{k}=\sin\theta_{k}\cos\phi_{k}$ for simplification.

For the NPA, the received signal can be modeled as [15]
(1)X=AS+N
where $S={\left[s_{1},s_{2},\cdots ,s_{K}\right]}^{T}\in C^{K\times L}$ is the signal matrix, $S_{k}$ represents the source vector and $s_{k}={\left[s_{k}(1),s_{k}(2),\cdots ,s_{k}(L)\right]}^{T}\in C^{L\times 1}$, L indicates the number of snapshots, A denotes the direction matrix of the array, and $A=[a_{y}(u_{1})⊗a_{x}(v_{1}),\cdots ,a_{y}(u_{K})⊗a_{x}(v_{K})]\in C^{{\left(M_{1}+M_{2}\right)}^{2}\times K}$, $a_{x}(v_{k})$ and $a_{y}(u_{k})$ denote the direction vectors along the *x*-axis and *y*-axis, respectively. They are indicated as $a_{y}(u_{k})={\left[1,e^{j2\pi d_{1}u_{k}/\lambda },\cdots ,e^{j2\pi M_{1}d_{1}u_{k}/\lambda },e^{j2\pi (M_{1}d_{1}+d_{2})u_{k}/\lambda },\cdots ,e^{j2\pi [M_{1}d_{1}+(M_{2}-1)d_{2}]u_{k}/\lambda }\right]}^{T}\in C^{(M_{1}+M_{2})\times 1}$ and $a_{x}(v_{k})={\left[1,e^{j2\pi d_{1}v_{k}/\lambda },\cdots ,e^{j2\pi M_{1}d_{1}u_{k}/\lambda },e^{j2\pi (M_{1}d_{1}+d_{2})u_{k}/\lambda },\cdots ,e^{j2\pi [M_{1}d_{1}+(M_{2}-1)d_{2}]u_{k}/\lambda }\right]}^{T}\in C^{(M_{1}+M_{2})\times 1}$. N denotes the received noise of the array, which is white Gaussian noise with zero mean and variance $\sigma^{2}$.

Actually, the covariance matrix of the received signal X with *L* snapshots can be approximated by
(2)$$\hat{R}=\frac{1}{L}XX^{H}$$

The Eigenvalue decomposition (EVD) of the covariance matrix $\hat{R}$ is expressed by
(3)$$\hat{R}={\hat{E}}_{s}{\hat{\Lambda }}_{s}{\hat{E}}_{s}^{H}+{\hat{E}}_{n}{\hat{\Lambda }}_{n}{\hat{E}}_{n}^{H}$$
where ${\hat{\Lambda }}_{s}$ and ${\hat{\Lambda }}_{n}$ are diagonal matrixes where the diagonal elements are denoted by the K largest Eigenvalues and (T−K) smallest Eigenvalues of $\hat{R}$, respectively. Signal subspace ${\hat{E}}_{s}$ contains the Eigenvectors of the K largest Eigenvalues and noise subspace ${\hat{E}}_{n}$ comprises the (T−K) smallest remaining Eigenvectors.

## 3. The Proposed Algorithm

In this section, different from other methods on NPAs that require virtualization with high computational complexity, we first transform the 2D spectral search into two 1D spectral search problems. Then, we exploit the 1D polynomial root finding method to estimate the two parameters. Finally, after completing the parameter matching, we can estimate the values of elevation and azimuth to complete DOA estimation.

### 3.1. D-MUSIC Spectrum Function

According to the orthogonal property between the signal steering vectors and the noise subspace, we form the spectrum function of the NPA as [29,32]
(4)$$P(u,v)=\frac{1}{a^{H}(u,v)E_{n}E_{n}^{H}a(u,v)}$$
where $a(u,v)=a_{y}(u)⊗a_{x}(v)\in C^{{\left(M_{1}+M_{2}\right)}^{2}\times 1}$ represents the direction vector of NPA, $a_{y}(u)={\left[1,e^{j2\pi d_{1}u/\lambda },\cdots ,e^{j2\pi M_{1}d_{1}u/\lambda },e^{j2\pi (M_{1}d_{1}+d_{2})u/\lambda },\cdots ,e^{j2\pi [M_{1}d_{1}+(M_{2}-1)d_{2}]u/\lambda }\right]}^{T}$, $a_{x}(v)={\left[1,e^{j2\pi d_{1}v/\lambda },\cdots ,e^{j2\pi M_{1}d_{1}v/\lambda },e^{j2\pi (M_{1}d_{1}+d_{2})v/\lambda },\cdots ,e^{j2\pi [M_{1}d_{1}+(M_{2}-1)d_{2}]v/\lambda }\right]}^{T}$, and $E_{n}$ is the noise subspace of NPA.

The DOA estimation can be obtained by a 2D spatial spectral search according to (4). Although it can be paired automatically, it requires expensive computation. To overcome this problem, we transform the 2D spectral search into two 1D spectral search problems, and then estimate the two parameters by exploiting the 1D polynomial root finding method.

### 3.2. Root-Finding Process of Reduced-Dimensional Polynomial

Construct polynomials based on the spectrum function (4) as
(5)$$\begin{array}{l} V(u,v)={\left[a_{y}(u)⊗a_{x}(v)\right]}^{H}E_{n}E_{n}^{H}[a_{y}(u)⊗a_{x}(v)]\\ =a_{x}^{H}(v){\left[a_{y}(u)⊗I_{M_{1}+M_{2}}\right]}^{H}E_{n}E_{n}^{H}[a_{y}(u)⊗I_{M_{1}+M_{2}}]a_{x}(v)\\ =a_{x}^{H}(v)Q(u)a_{x}(v) \end{array}$$
and
(6)$$\begin{array}{l} V(u,v)={\left[a_{y}(u)⊗a_{x}(v)\right]}^{H}E_{n}E_{n}^{H}[a_{y}(u)⊗a_{x}(v)]\\ =a_{y}^{H}(u){\left[I_{M_{1}+M_{2}}⊗a_{x}(v)\right]}^{H}E_{n}E_{n}^{H}\left[I_{M_{1}+M_{2}}⊗a_{x}(v)\right]a_{y}(u)\\ =a_{y}^{H}(u)Q(v)a_{y}(u) \end{array}$$
where $Q(u)={\left[a_{y}(u)⊗I_{M_{1}+M_{2}}\right]}^{H}E_{n}E_{n}^{H}[a_{y}(u)⊗I_{M_{1}+M_{2}}]$ and $Q(v)={\left[I_{M_{1}+M_{2}}⊗a_{x}(v)\right]}^{H}E_{n}E_{n}^{H}[I_{M_{1}+M_{2}}⊗a_{x}(v)]$.

According to the rank relationship of the matrix product, the condition below must be met
(7)$$0<Rank(E_{n}E_{n}^{H})\leq Rank(E_{n})$$
and then we have
(8)$$\det\left\{E_{n}E_{n}^{H}\right\}\neq 0$$

Obviously, on the basis of (8), we can conclude that det{Q(u)} is a nonzero polynomial; hence, Q(u) is a factor of
V(u,v).
As Q(u)
is only related to u, the roots of det{Q(u)}=0 can establish the following formula:(9)$$V(u,v)=a_{x}^{H}(v)Q(u)a_{x}(v)=0$$

The estimates of u and v can be expressed by the problem of 1D polynomial root finding. Therefore, the pairwise estimates of u and v obtained from 2D spectral research are changed into 1D root finding processes. After that, reconstruct (5) and (6) as
(10)$$\det\left\{Q(u)\right\}=\det\left\{{\left[a_{y}(u)⊗I_{M_{1}+M_{2}}\right]}^{H}E_{n}E_{n}^{H}[a_{y}(u)⊗I_{M_{1}+M_{2}}]\right\}=0$$
(11)$$\det\left\{Q(v)\right\}=\det\left\{{\left[I_{M_{1}+M_{2}}⊗a_{x}(v)\right]}^{H}E_{n}E_{n}^{H}[I_{M_{1}+M_{2}}⊗a_{x}(v)]\right\}=0$$

Define
(12)$$\begin{array}{l} z_{1}=e^{j2\pi du/\lambda }\\ z_{2}=e^{j2\pi dv/\lambda } \end{array}$$
where d=λ/2.

Define the direction vector along the *x*-axis for the UPA with the same array aperture size as the NPA: 
$a_{Ex}(v)={\left[1,e^{j2\pi dv/\lambda },\cdots ,e^{j2\pi (\min\{M_{1},M_{2}\}+1)\max\{M_{1},M_{2}\}-1)dv/\lambda }\right]}^{T}\in C^{((\min\{M_{1},M_{2}\}+1)\max\{M_{1},M_{2}\})\times 1}$ For simplicity, we assume that
$M_{1}\leq M_{2}$
and that
$a_{Ex}(v)$
is represented as
$a_{Ex}(v)={\left[1,e^{j2\pi dv/\lambda },\cdots ,e^{j2\pi \left((M_{1}+1)M_{2}-1\right)dv/\lambda }\right]}^{T}\in C^{((M_{1}+1)M_{2})\times 1}$
Due to the same array aperture between the NPA and UPA, we have
(13)$$a_{x}(v)=Ga_{Ex}(v)$$
where $G\in Z^{\left(M_{1}+M_{2})\times ((M_{1}+1)M_{2}\right)}$. Specifically, $g_{ij}=1$ holds when the ith sensor in the $a_{x}(v)$ overlaps with the jth sensor in the $a_{Ex}(v)$, or else $g_{ij}=0$, where $g_{ij}$ is the (i,j)th element of G. For the NPA shown in Figure 1, G is a 5×9 matrix, in which four columns are zeros. Furthermore, we can also obtain the relation $a_{y}(u)=Ga_{Ey}(u)$.

Accordingly, the direction vectors can be further expressed as
(14)$$a_{y}(u)=Ga_{Ey}(u)=G{\left[1,e^{j2\pi du/\lambda },\cdots ,e^{j2\pi \left((M_{1}+1)M_{2}-1\right)du/\lambda }\right]}^{T}=G{\left[1,z_{1},\cdots ,z_{1}^{\left((M_{1}+1)M_{2}-1\right)}\right]}^{T}=a_{y}(z_{1})$$
(15)$$a_{x}(v)=Ga_{Ex}(v)=G{\left[1,e^{j2\pi dv/\lambda },\cdots ,e^{j2\pi \left((M_{1}+1)M_{2}-1\right)dv/\lambda }\right]}^{T}=G{\left[1,z_{2},\cdots ,z_{2}^{\left((M_{1}+1)M_{2}-1\right)}\right]}^{T}=a_{x}(z_{2})$$

Without losing generality, we can substitute $z_{1}^{(M_{1}+1)M_{2}-1}{\left[a_{y}^{T}(z_{1}^{-1})⊗I_{M_{1}+M_{2}}\right]}^{H}$ for ${\left[a_{y}(u)⊗I_{M_{1}+M_{2}}\right]}^{H}$ and substitute $z_{2}^{(M_{1}+1)M_{2}-1}{\left[I_{M_{1}+M_{2}}⊗a_{x}^{T}(z_{2}^{-1})\right]}^{H}$ for ${\left[I_{M_{1}+M_{2}}⊗a_{x}(v)\right]}^{H}$. Namely,
(16)$$\det\left\{Q(z_{1})\right\}=\det\left\{z_{1}^{\left((M_{1}+1)M_{2}-1\right)}{\left[a_{y}^{T}(z_{1}^{-1})⊗I_{M_{1}+M_{2}}\right]}^{H}E_{n}E_{n}^{H}[a_{y}(z_{1})⊗I_{M_{1}+M_{2}}]\right\}=0$$
(17)$$\det\left\{Q(z_{2})\right\}=\det\left\{z_{2}^{\left((M_{1}+1)M_{2}-1\right)}{\left[I_{M_{1}+M_{2}}⊗a_{x}^{T}(z_{2}^{-1})\right]}^{H}E_{n}E_{n}^{H}[I_{M_{1}+M_{2}}⊗a_{x}(z_{2})]\right\}=0$$

According to the above formula, ${\hat{u}}_{k}$ and ${\hat{v}}_{i}$ are acquired from the K-root distribution nearest to the unit circle based on (16) and (17), which are expressed as ${\hat{z}}_{11},\cdots ,{\hat{z}}_{1k},\cdots ,{\hat{z}}_{1K}$ and ${\hat{z}}_{21},\cdots ,{\hat{z}}_{2k},\cdots ,{\hat{z}}_{2K}$, respectively, i.e.,
(18)$${\hat{u}}_{k}=\left(\frac{angle({\hat{z}}_{1k})\lambda }{2\pi d}\right),k=1,\cdots ,K$$
(19)$${\hat{v}}_{i}=\left(\frac{angle({\hat{z}}_{2i})\lambda }{2\pi d}\right),i=1,\cdots ,K$$

Among the traditional DOA estimation methods with sparse arrays, the array element separation of the two subarrays is larger than half wavelength; as a result, it is necessary to eliminate the ambiguity. According to the structure of the NPA, since it has at least one set of adjacent sensors spaced no more than half a wavelength apart, it is an unambiguous array and can avoid additional disambiguation operations.

Since ${\hat{u}}_{k}$ and ${\hat{v}}_{i}$ in the above operation are obtained by finding the roots of two 1D polynomials, they are not paired with each other, so we need to pair these unpaired parameters. After the parameters are paired, we can directly obtain the real DOA estimation.

### 3.3. Parament Pairing and DOA Estimation

Since the two root-finding processes of ${\hat{u}}_{k}$ and ${\hat{v}}_{i}$ are carried out separately, in this part, we pair them up. Therefore, a cost function is given as
(20)$$V_{k,i}=\underset{i=1,\cdots ,K}{\arg}\min\left\|a^{H}({\hat{u}}_{k},{\hat{v}}_{i})E_{n}E_{n}^{H}a({\hat{u}}_{k},{\hat{v}}_{i})\right\|,(k=1,\cdots ,K)$$
where $a({\hat{u}}_{k},{\hat{v}}_{i})$ denotes the direction vector constructed by ${\hat{u}}_{k}$ and ${\hat{v}}_{i}$, which can be obtained on the basis of (1).

For any ${\hat{u}}_{k}$, we can calculate the values of i and k when $V_{k,i}(1\leq i\leq K)$ takes the minimum value, and then we define this i paired with k as $i^{\prime }$. Finally, the elevation and azimuth angles can be calculated by
(21)$${\hat{\theta }}_{k}=arc\sin(abs({\hat{u}}_{k}+j{\hat{v}}_{i’})),1\leq k\leq K$$
(22)$${\hat{\phi }}_{k}=angle({\hat{u}}_{k}+j{\hat{v}}_{i’}),1\leq k\leq K$$

### 3.4. The Main Steps of the Proposed Algorithm

We provide a summary of the main steps of the proposed algorithm below:

Step 1: Calculate the covariance matrix $\hat{R}$ of the received signal X; the noise subspace $E_{n}$ is obtained by EVD;

Step 2: Reduce the dimension of constructed polynomial V(u,v) according to (4);

Step 3: Estimate ${\hat{u}}_{k}$ and ${\hat{v}}_{i}$ by the 1D polynomial root finding method;

Step 4: Match the parameters ${\hat{u}}_{k}$ and ${\hat{v}}_{i}$ based on (20);

Step 5: Calculate ${\hat{\theta }}_{k}$ and ${\hat{\phi }}_{k}$ on the basis of (21) and (22).

## 4. Performance Analysis

### 4.1. Complexity Analysis

In this part, we compare the computational complexity of the proposed algorithm with other algorithms, including the ESPRIT algorithm for NPAs [34] and some other MUSIC based algorithms such as the 2D-PSS [15] and RD-ROOT-UPA [23] methods for UPAs. For the proposed algorithm, it takes $O\left\{T^{2}L\right\}$ to calculate the covariance matrix and the computational complexity of EVD is $O\left\{T^{3}\right\}$. The polynomial root-finding process needs $O\left\{2{\left(2{\left(\left(M_{1}+1\right)M_{2}\right)}^{2}\right)}^{3}\right\}$ and the parameter matching process costs $O\left\{K^{2}\left(T-K\right)\left(T+1\right)\right\}$. Therefore, the total complexity is $O\left\{T^{2}L+T^{3}+2{\left(2{\left(\left(M_{1}+1\right)M_{2}\right)}^{2}\right)}^{3}+K^{2}\left(T-K\right)\left(T+1\right)\right\}$.

According to the computational complexity of the above algorithms listed in Table 1 in which Δ = 0.0001 is the spectral search interval, *M* = *M*_1_ + *M*_2_ represents the number of elements in each row and column of the uniform planar array (UPA). Figure 2 shows the computational complexity of different methods with the increase of sensors, where *K* = 2 and *L* = 200. It can be observed from Figure 2 that the complexity of the proposed algorithm is significantly lower than that of 2D-PSS, because the computational efficiency of polynomial root finding is much higher than that of spectral search. Meanwhile, it can be observed that the complexity of the proposed algorithm is similar to that of the ESPRIT algorithm, but higher accuracy can be obtained by finding the root of the spectral function. Finally, although the complexity of the proposed algorithm is higher than that of the RD-ROOT-UPA algorithm, the larger array aperture of the NPA compared to the UPA gives better estimation accuracy with an equivalent number of sensors.

### 4.2. Cramer-Rao Bound

The Cramer–Rao bound (CRB) is a lower bound of the error variance of a parameter estimator. In this part, we derive the CRB of 2D DOA estimation for NPAs.

According to [35], the CRB of an NPA can be given by
(23)CRB=σ22L{Re[DHΠA⊥D⊕P^T]}−1
where $A=\left[\begin{array}{l} A_{1}\\ A_{2} \end{array}\right]$, $A_{i}=[a_{yi}(\theta_{1},\phi_{1})⊗a_{xi}(\theta_{1},\phi_{1}),\cdots ,a_{yi}(\theta_{K},\phi_{K})⊗a_{xi}(\theta_{K},\phi_{K})](i=1,2)$, $D=[\partial a_{1}/\partial \theta_{1},\cdots ,\partial a_{K}/\partial \theta_{K},\partial a_{1}/\partial \phi_{1},\cdots ,\partial a_{K}/\partial \phi_{K}]$, $\Pi_{A}^{⊥}=I_{M_{1}^{2}+M_{2}^{2}}-A{\left[A^{H}A\right]}^{-1}A^{H}$, $\hat{P}=\left[\begin{array}{cc} {\hat{P}}_{S} & {\hat{P}}_{S}\\ {\hat{P}}_{S} & {\hat{P}}_{S} \end{array}\right]$; ${\hat{P}}_{S}=SS^{H}/L$, $a_{k}$ represents the kth column of A, and the variance of the received noise is $\sigma^{2}$.

### 4.3. Achievable DOFs

For the traditional 2D DOA estimation algorithms of a UPA, such as 2D-PSS, the DOF is $M^{2}-1$, where $M=M_{1}+M_{2}$ represents the number of elements in each row and column of the UPA. According to [32], the RD-ROOT-UPA algorithm can distinguish M(M−1) signals at most, which reduces the dimension and directly finds the roots of the spectral function of the UPA. The proposed algorithm utilizes an RD root finding technique to reduce the computational complexity, and we can obtain the achievable DOFs on the basis of (7) as follows:(24)$$1\leq Rank(E_{n}E_{n}^{H})\leq Rank(E_{n})$$
where $Rank(E_{n})=\min\left\{{\left(M_{1}+M_{2}\right)}^{2},{\left(M_{1}+M_{2}\right)}^{2}-K\right\}={\left(M_{1}+M_{2}\right)}^{2}-K$. Therefore, the proposed algorithm can identify the maximum number of signals as
(25)$$K\leq {\left(M_{1}+M_{2}\right)}^{2}-1$$

We provide the achievable DOFs of different methods in Table 2. Compared with the traditional 2D DOA estimation methods, it is clear that the proposed algorithm is able to achieve relatively high DOFs.

### 4.4. Advantages

On account of the above analysis, the main advantages of the proposed algorithm are summarized as below:
The proposed algorithm transforms the problem of 2D spectral search into two 1D polynomial root finding problems, which greatly reduces the complexity compared with the traditional 2D-MUSIC algorithms such as the 2D-PSS algorithm;Compared with other 2D DOA estimation method for NPAs, the proposed algorithm avoids virtualization steps and spatial smoothing, so it has lower complexity than ESPRIT-NPA;The proposed algorithm fully utilizes the expanded aperture and DOF of the NPA, allowing it to achieve a higher DOF and array aperture compared to RD-ROOT-UPA and 2D-PSS algorithms based on uniform arrays.


## 5. Simulations

In this section, assuming that there are *K* uncorrelated far-field narrowband signals impinging on the NPA, we provide extensive simulations to verify the performance of the proposed algorithm in terms of the root mean square error (RMSE) under various conditions. According to [33], the RMSE is expressed as follows:(26)$$RMSE=\frac{1}{K}\sum_{k=1}^{K}\sqrt{\frac{1}{C}\sum_{i=1}^{C}{\left({\hat{\phi }}_{k,i}-\phi_{k}\right)}^{2}+{\left({\hat{\theta }}_{k,i}-\theta_{k}\right)}^{2}}$$
where C=500 is the number of trials; ${\hat{\phi }}_{k,i}$ and ${\hat{\theta }}_{k,i}$ are the estimated values of the kth signal in the ith test that correspond with the true $\phi_{k}$ and $\theta_{k}$, respectively.

### 5.1. Scatter Figure

In this part, we draw the scatter figure of *K* sources estimated by the proposed algorithm in Figure 3, where
SNR=15 dB,
K=6,
L=400
are the snapshots; the *K* sources are
$\left(\theta_{1},\phi_{1})=(5^{\circ },10^{\circ }\right)$,
$\left(\theta_{2},\phi_{2})=(15^{\circ },20^{\circ }\right)$,
$\left(\theta_{3},\phi_{3})=(25^{\circ },30^{\circ }\right)$,
$\left(\theta_{4},\phi_{4})=(35^{\circ },40^{\circ }\right)$,
$\left(\theta_{5},\phi_{5})=(45^{\circ },50^{\circ }\right)$
and
$\left(\theta_{6},\phi_{6})=(55^{\circ },60^{\circ }\right)$
respectively. The NPA is a planar array extended configuration of a two-level NLA with
$M_{1}=2$
and
$M_{2}=2$
. As depicted in Figure 3, the proposed algorithm is able to estimate all sources on the NPA accurately.

### 5.2. RMSE Results of the Proposed Algorithm Versus Snapshots

Here, we present the RMSE result of the proposed algorithm with different snapshots in Figure 4, where $\left(\theta_{1},\phi_{1})=(20^{\circ },30^{\circ }\right)$, $\left(\theta_{2},\phi_{2})=(40^{\circ },50^{\circ }\right)$, $M_{1}=2$ and $M_{2}=2$. The results suggest that due to the increasingly accurate covariance estimation results, the estimation accuracy increases with the increase of the snapshots.

### 5.3. RMSE Results of the Proposed Algorithm Versus Number of Sensors

Herein, Figure 5 shows the relationship between the RMSE results of the proposed algorithm and the number of sensors, where $\left(\theta_{1},\phi_{1})=(20^{\circ },30^{\circ }\right)$, $\left(\theta_{2},\phi_{2})=(40^{\circ },50^{\circ }\right)$, and L=200. As depicted in Figure 5, with the number of array elements increasing, the DOA estimation performance improves because of diversity gain of the receiving antenna increases.

### 5.4. RMSE Comparison of Different Algorithms

Figure 6 and Figure 7 exhibit the comparison of RMSE results of different algorithms when snapshots and SNR change separately, including the proposed algorithm, RD-ROOT-UPA algorithm, 2D-PSS algorithm, and ESPRIT-NPA algorithm, where $M_{1}=2$ and $M_{2}=2$ is the structural configuration of used NPA. In order to ensure the same total number of array elements, the UPA used for the RD-ROOT-UPA and 2D-PSS algorithms is set to M×M=4×4.

Figure 6 shows that all the algorithms obtain better estimation results with the increase of snapshots; when the SNR is 10 dB, the performance of the 2D-PSS algorithm is better than that of RD-ROOT-UPA and ESPRIT-NPA because the performance of spectral peak search will be better than that of spectral function root finding method. It will also be better than the ESPRIT method based on rotation invariance. The proposed algorithm achieves the best estimation performance because the proposed method can exploit the extended array aperture and increased DOF offered by NPAs.

The result of Figure 7 shows that the proposed algorithm and ESPRRIT-NPA algorithm outperform other algorithms for UPAs when the SNR is low, profiting from the larger array aperture of NPA, and the proposed algorithm performs two root finding processes directly according to the Formula (4), avoiding the performance degradation caused by intermediate operations. The 2D-PSS algorithm obtains better performance with spectral peak search than the RD-ROOT-UPA algorithm which directly finds the root of the same uniform array. As the SNR increases gradually, the performance of ESPRRIT-NPA becomes worse than that of other algorithms, but the proposed algorithm is still the best.

## 6. Conclusions

In this paper, we proposed a 2D DOA estimation method for NPAs with high computational efficiency and no virtualization by using the root finding method of reduced dimension polynomials. The proposed algorithm converts the 2D spectral peak search into two 1D polynomial root finding problems, and then after parameters pairing, 2D DOA estimation is realized, which is directly processed by the root-finding MUSIC method, avoiding the process of virtualization and spatial smoothing as well as the huge computational load brought by spectral peak search while maintaining estimation accuracy. Simulation results verified that the proposed method outperforms many exiting methods in terms of computational complexity and estimation accuracy. In future research, we will consider using this dimensionality reduction and root-finding method on improved arrays with larger array apertures, or consider combining this method with virtual array technology to obtain greater DOFs.

## Figures and Tables

**Figure 1 sensors-22-05220-f001:**
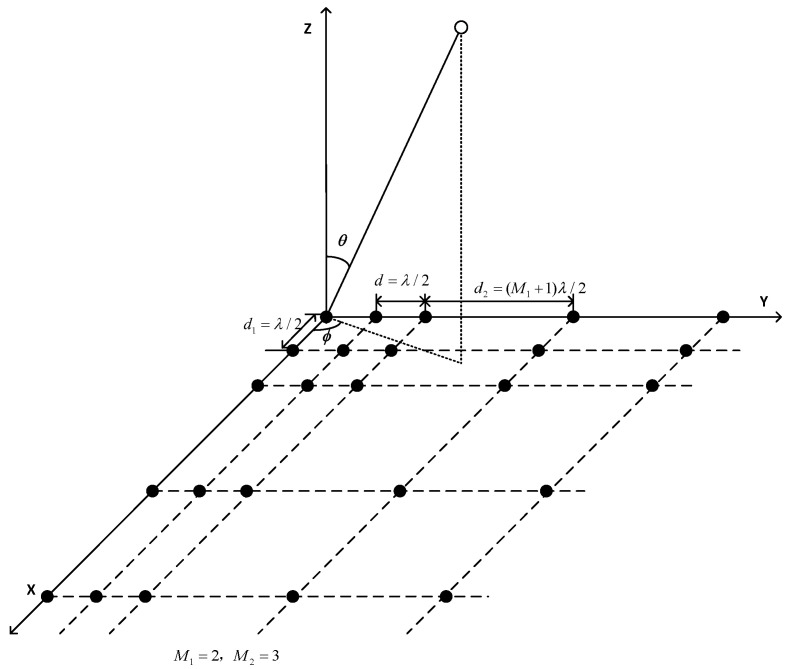
The structural layout of an NPA with $M_{1}=2$ and $M_{2}=3$.

**Figure 2 sensors-22-05220-f002:**
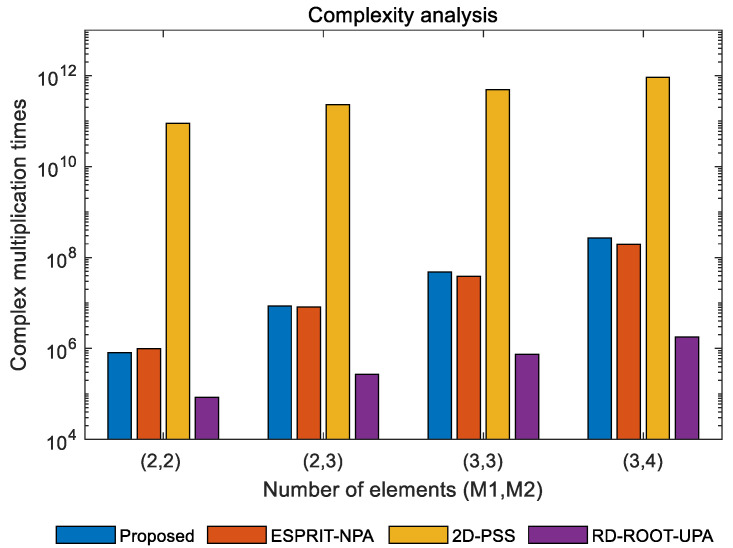
Comparison of complexity with different numbers of elements.

**Figure 3 sensors-22-05220-f003:**
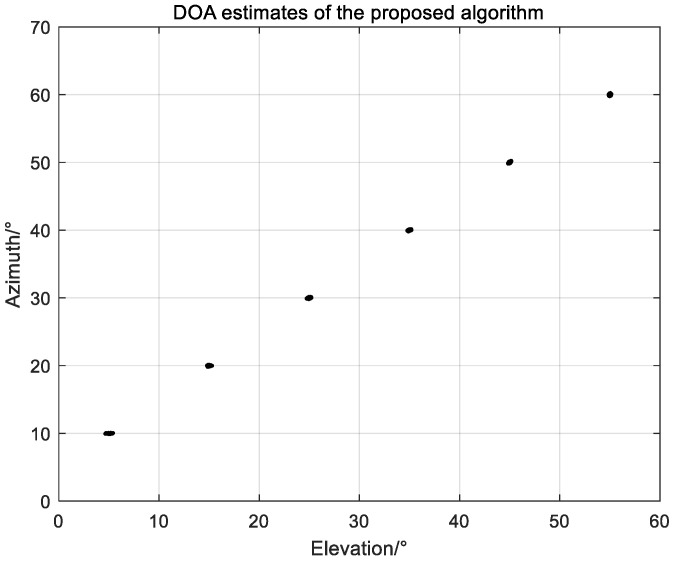
DOA estimates of the proposed algorithm.

**Figure 4 sensors-22-05220-f004:**
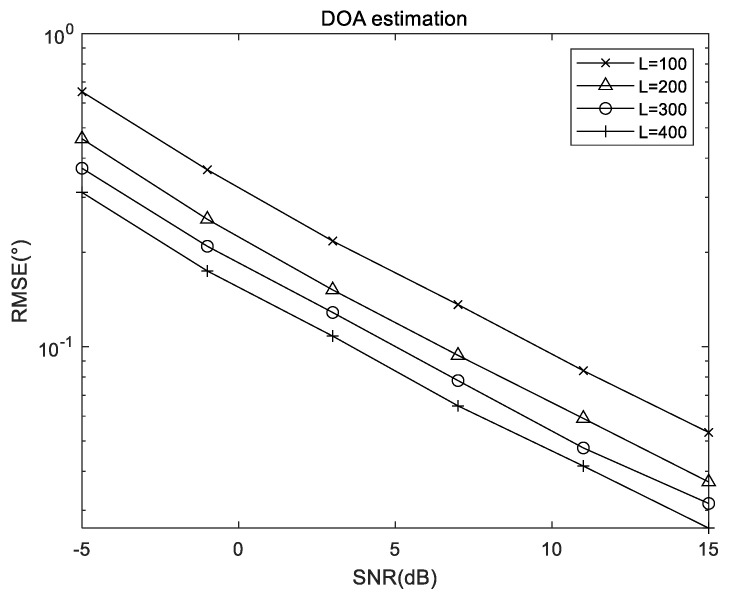
RMSE performance of the proposed algorithm versus snapshots.

**Figure 5 sensors-22-05220-f005:**
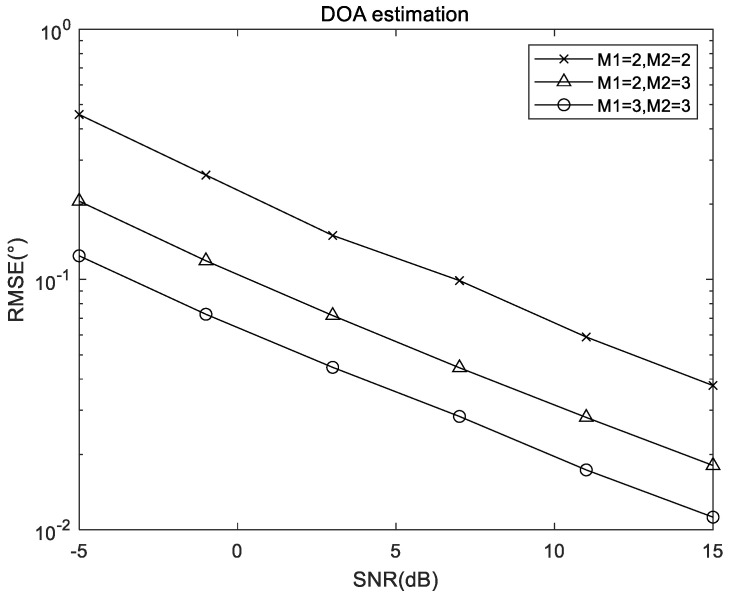
RMSE performance of the proposed algorithm versus number of sensors.

**Figure 6 sensors-22-05220-f006:**
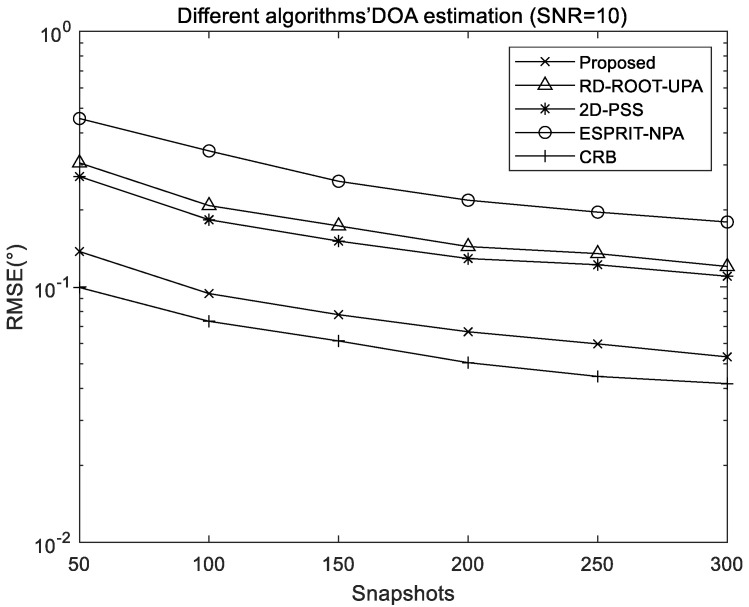
RMSE comparison of different algorithms with different snapshots.

**Figure 7 sensors-22-05220-f007:**
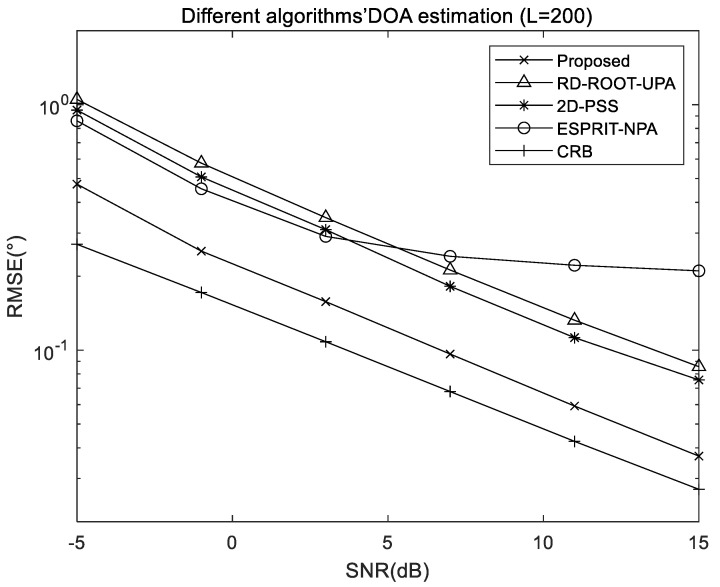
RMSE comparison of different algorithms with different SNR.

**Table 1 sensors-22-05220-t001:** Complexity of different methods.

Algorithm	Computational Complexity
Proposed	$O\left\{T^{2}L+T^{3}+2{\left(2{\left((M_{1}+1)M_{2}\right)}^{2}\right)}^{3}+K^{2}(T-K)(T+1)\right\}$
ESPRIT-NPA	$O\{{\left({\left(M_{1}+M_{2}\right)}^{2}\right)}^{2}L+{\left({\left((M_{1}+1)M_{2}+1\right)}^{2}\right)}^{3}$ $+4K^{2}((M_{1}+1)M_{2}+1)((M_{1}+1)M_{2})+6K^{3}\}$
2D-PSS	$O\left\{T^{2}L+T^{3}+4T(T-K)/(\Delta^{2})\right\}$
RD-ROOT-UPA	$O\left\{M^{4}L+M^{6}+2{\left(2M(M-1)\right)}^{3}+K^{2}(M^{2}-K)(M^{2}+1)\right\}$

**Table 2 sensors-22-05220-t002:** Achievable DOFs of different methods.

Algorithm	Achievable DOFs
Proposed	${\left(M_{1}+M_{2}\right)}^{2}-1$
2D-PSS	$M^{2}-1$
ESPRIT-NPA	${\left((M_{1}+1)M_{2}+1\right)}^{2}-1$
RD-ROOT-UPA	M(M−1)

## Data Availability

Not applicable.

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
