# Peer review of "A Computationally Efficient and Virtualization-Free Two-Dimensional DOA Estimation Method for Nested Planar Array: RD-Root-MUSIC Algorithm"

_sensors, 2022, doi:10.3390/s22145220_

Round 1

Reviewer 1 Report

This paper presents a computationally efficient 2D DOA estimation method for nested planar arrays, which decouples the 2D problem into two 1D ones, and  finally paired to achieve 2D DOA estimation. The performance improvement is also verified via extensitive simulatoins. In general, I just have several minor concerns,

(1) Some symboles are missing, e.g., in Lines 111, 120, 122, 154, etc. 

(2) In simulations, what is the array geometry for 2D PSS? 

(3) If possible, the automatic pairing of the 2D angels should be considered.

(4) The figures and tables are in correct format and can prove the improvement of the proposed method.

Author Response

Dear Reviewers,
Thank you for your Email, and many thanks for coordination with our paper. The authors are grateful for comments and helps from the editor and reviewers. We also cherish this valuable revision opportunity by further improving the quality of our manuscript based on your suggestions, including technical aspects, writing and presentation. The current manuscript has been revised and refreshed by our co-authors. According to the comments of editor and reviewers, I hereby write a revision report and list the modifications, please see the attachment.

Reviewer 2 Report

In this paper, the authors propose an efficient 2D DOA estimation method by using the reduced dimension and root-finding MUSIC algorithm with nested planar array. Simulation results are provided to verify the advantage of the proposed algorithm. Overall, the study is both complete and convincing. I have the following concerns:

1. It should attach great importance to the writing standard of this paper, such as the missing symbol.

Line 119: There is no space to the left of the word “where”. Please correct these problems in other places.

2. Introduction: Lots of newly DOA estimation method based on information geometry [1-2]. More knowledge about information geometry can be referred to [3]. It’s better to cite these references.

[1] Y. Dong, C. Dong, W. Liu, M. Liu and Z. Tang, "Scaling Transform Based Information Geometry Method for DOA Estimation," in IEEE Transactions on Aerospace and Electronic Systems, vol. 55, no. 6, pp. 3640-3650, Dec. 2019, doi: 10.1109/TAES.2019.2910363.

[2] M. Coutino, R. Pribic and G. Leus, "Direction of arrival estimation based on information geometry," 2016 IEEE International Conference on Acoustics, Speech and Signal Processing (ICASSP), 2016, pp. 3066-3070, doi: 10.1109/ICASSP.2016.7472241.

[3] X. Hua, Y. Ono, L. Peng and Y. Xu, "Unsupervised Learning Discriminative MIG Detectors in Nonhomogeneous Clutter," in IEEE Transactions on Communications, vol. 70, no. 6, pp. 4107-4120, June 2022, doi: 10.1109/TCOMM.2022.3170988.

3. More comparison results should be provided to validate the superiority of the proposed method. Please provide the complexity analysis of the proposed algorithm with respect to the compared algorithms. It's better to validate your efficiency in terms of theoretical and simulation analysis.

4. Conclusion: The authors missed the further improvements and the potential drawbacks of the proposed method. Make sure your conclusions reflect on the strengths and weaknesses of your work, how others in the field can benefit from it and thoroughly discuss future work.

Author Response

(The authors gave the same response as above.)

Round 2

Reviewer 2 Report

The authors have addressed all my concerns.